# AdaptFormer: Adapting Vision Transformers for Scalable Visual Recognition

**Shoufa Chen**[1*]  **Chongjian Ge**[1*]  **Zhan Tong**[2]  **Jiangliu Wang**[2]
**Yibing Song**[2]  **Jue Wang**[2]  **Ping Luo**[1]

[1]The University of Hong Kong  [2]Tencent AI Lab

## Abstract

Pretraining Vision Transformers (ViTs) has achieved great success in visual recognition. A following scenario is to adapt a ViT to various image and video recognition tasks. The adaptation is challenging because of heavy computation and memory storage. Each model needs an independent and complete finetuning process to adapt to different tasks, which limits its transferability to different visual domains. To address this challenge, we propose an effective adaptation approach for Transformer, namely AdaptFormer, which can adapt the pre-trained ViTs into many different image and video tasks efficiently. It possesses several benefits more appealing than prior arts. Firstly, AdaptFormer introduces lightweight modules that only add less than 2% extra parameters to a ViT, while it is able to increase the ViT's transferability without updating its original pre-trained parameters, significantly outperforming the existing 100% fully fine-tuned models on action recognition benchmarks. Secondly, it can be plug-and-play in different Transformers and scalable to many visual tasks. Thirdly, extensive experiments on five image and video datasets show that AdaptFormer largely improves ViTs in the target domains. For example, when updating just 1.5% extra parameters, it achieves about 10% and 19% relative improvement compared to the fully fine-tuned models on Something-Something v2 and HMDB51, respectively. Code is available at https://github.com/ShoufaChen/AdaptFormer.

## 1  Introduction

There is a growing interest in adopting a general neural model to tackle a large variety of different tasks since it benefits in reducing the need for task-specific model design and training. Recently, Transformer [81] demonstrates great potential in this goal considering its success in various fields, *e.g.*, natural language processing (NLP) [27, 10, 82, 88], visual recognition [31, 79, 90, 63], dense prediction [83, 11, 98, 96, 86], Generative Adversarial Network (GAN) [52, 48], reinforcement learning (RL) [18, 16, 87], robotics [50, 25], and etc. However, existing literature in computer vision tend to focus on the *same network with task-specific weights* scenario, where a single network is used to train from scratch or fully fine-tune on a specific dataset, making it infeasible to maintain a separate model weight for every dataset when the number of task grows, especially for the increasing model capacity of state-of-the-art models (*e.g.*, ViT-G/14 [93] with over 1.8 billion parameters).

Different from prior arts, we step into the direction of developing *same network with almost same weights* and achieve superior performance than the full-tuning approach by only tuning less than 2% parameters, with the remaining over 98% parameters shared across different tasks. There are two challenges to learning universal representations using a single model. The first one lies in the pre-training stage, which requires algorithms that can learn well-generalized representations that are

---

*Equal contribution.

easy to be applied to many tasks. Recent arts in self-supervised learning [12, 5, 43, 97, 85, 78, 35] can serve as a solution to this challenge. The second one, which is our main concern in this work, is to build an effective pipeline that can adapt the model obtained at the pre-training stage to various downstream tasks by tuning parameters as less as possible and keeping the left parameters frozen.

While fine-tuning pre-trained models has been widely studied in NLP [6, 46, 69, 70, 58, 56, 47, 92, 62, 42], this topic is seldomly explored in the vision, where full-tuning of model parameters is still the dominant strategy for adapting vision transformers. However, the full fine-tuning cannot satisfy the goal of *universal representation* as it assigns an independent set of weights for every task. Linear probing is a straightforward approach to maintaining the pre-trained model fixed by only tuning a specific lightweight classification head for every task. However, linear probing tends to have an unsatisfactory performance and misses the opportunity of pursuing strong but non-linear features [43], which indeed benefit deep learning. More recently, Bahng *et.al.,* [4] aimed to adapt pre-trained models by modifying raw input pixel space. Jia *et.al.,* [51] proposed Visual Prompt Tuning (VPT) to adapt transformer models for downstream vision tasks, which prepends several learnable parameters (*prompts*) to the patch embeddings and freezes the whole pre-trained backbone.

In this work, we propose a lightweight module, namely AdaptFormer, to adapt vision transformers by updating the weights of Adapt-Former. We introduce learnable parameters from the model perspective, which is different from VPT, which inserts learnable parameters into the token space. Our AdaptFormer is conceptually simple yet effective. It consists of two fully connected layers, a non-linear activation function, and a scaling factor. This module is set in parallel to the feed-forward network (FFN) of the original ViT model, as shown in Figure 2b. This design is turned out to be effective for model transfer when processing scalable visual tokens for both image and video data (i.e., image data consists of a small scale of visual tokens while video data consists of a large scale). As shown in Figure 1, compared with the full-tuning strategy, AdaptFormer achieves comparable per-

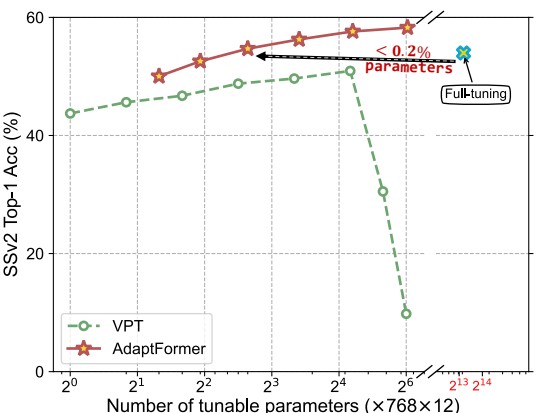

Figure 1: **Parameter-Accuracy trade-off.** We leverage ViT-Base as backbone and report top-1 accuracy on SSv2 dataset. AdaptFormer can surpass full-tuning with only 0.2% tunable parameters. More detailed results are shown in Table 1.

formance on video recognition with only about 0.1% tunable parameters. Meanwhile, with less than 2% tunable parameters, AdaptFormer surpasses the full-tuning solution by about 10% on top-1 accuracy. Similar approaches are also proposed in fine-tuning pre-trained language models (PLMs) [6, 46, 70, 42].

The key **contributions** of this paper are summarized as follows: **(1)** We propose a simple yet effective framework, namely AdaptFormer, for adapting vision transformers to a large variety of downstream visual recognition tasks and avoiding catastrophic interference with each other. To the best of our knowledge, this is the first work that explores efficient fine-tuning in video action recognition. **(2)** We ablate many design choices and demonstrate the superior robustness of AdaptFormer when parameters scale up. **(3)** Extensive experiments on various downstream tasks demonstrate that AdaptFormer outperforms existing fine-tuning approaches significantly. By demonstrating the effectiveness of AdaptFormer on multiple visual benchmarks, we hope our work could inspire the research communities to rethink the fine-tuning mechanism in computer vision and make progress toward a flexible yet universal Transformer model for visual recognition.

## 2 Related Works

In the proposed AdaptFormer, we mainly introduce a plug-and-play module for efficiently fine-tuning the current vision Transformer models. In this section, we perform a literature review on related works from two perspectives, *i.e.*, the vision Transformers, and efficient transfer learning for vision Transformers.

## 2.1 Transformer in Vision

The Transformer architecture is first introduced in [81] and has re-energized the natural language processing (NLP) field from then on [27, 10]. Inspired by its huge success, researches in the computer vision filed have also evolved into Transformer era since ViTs [31]. The strong capability of modeling long-range relation has facilitated Transformer in various vision tasks, including image classification [31, 63, 60], object detection [11, 98, 22], semantic/instance segmentation [86], video understanding [8, 2, 33, 57], point cloud modeling [95, 41], 3D Object Recognition [20] and even low-level processing [17, 59, 84]. Furthermore, transformers have advanced the vision recognition performance by a large-scale pretraining [21, 67, 13, 36, 43, 78, 71]. In such a situation, given the pre-trained Transformer models, which are more larger than the previously prevalent CNN backbones, one open question is how to fine-tune the big vision models so that they can be adapted into more specific down-stream tasks. To solve the open question, we propose AdaptFormer to transfer ViTs from the pre-trained pre-texts into the target tasks in a more effective and efficient way.

## 2.2 Efficient Transfer learning for Transformers

Transfer learning targets re-adopting a pre-trained model (either via the supervised or the unsupervised manner) as the starting point and further fine-tuning the specific model on a new task. In the NLP field, transferring the large pre-trained language models (PLMs) [27, 10] into downstream tasks has been the popular paradigm for a long time. Conventional arts [27, 10] set all the network parameters as learnable ones and adapt them to the target tasks. However, with the growth of model sizes and the complexity of the specific tasks, the conventional paradigm is inevitably limited by the huge computational burden. The NLP community has explored several ways for parameter-efficient transfer learning that only set a few parameters learnable and fine-tune them for efficiency. The pioneer works could be mainly categorized from the token [58, 56] and network perspectives [46, 47, 92, 40]. Basically speaking, the token-related methods [56, 58] typically prepend several learnable prefix vectors/tokens to the projected tokens within the multi-head self-attention layers (MHSA [81]). The philosophy behind it is to assist the pre-trained models in understanding downstream tasks with the guidance of extra token information. On the other hand, network-related methods [46, 47] integrate shallow modules to improve the model transferability. The introduced modules adapt the produced representations into the downstream tasks via features fusion.

Recently, with the emergence of a much more large-scale dataset [26, 72, 74, 66, 53], increasing researchers in computer vision have adopted the homologous paradigm, *i.e.*, first pre-training and then fine-tuning, to advance the vision tasks. As for the second stage, traditional methods typically adopt the full-tuning arts in the downstream tasks. Rare attention has been drawn to the field of efficient adaptation, especially in the field of vision Transformers. Inspired by Prompting in NLP, [51] introduced the learnable tokens in exploring the efficient adaptation for ViTs. We empirically found that the performance of prompting is hindered by the scale of tokens. That is to say, for the tasks where the number of tokens is on a small scale, *e.g.*, image classification, Prompting is efficient for improving the model transferability. However, for larger scale tokens, *e.g.*, video understanding, Prompting presents limited potential. This observation motivates us to introduce AdaptFormer, which is effective in the scenarios of scalable visual tokens.

## 3 Approach

We propose AdaptFormer for efficiently transferring large pre-trained vision transformer models to downstream tasks, in both image and video domains. AdaptFormer attains strong transfer learning abilities by only fine-tuning a small number of extra parameters, circumventing catastrophic interference among tasks. We illustrate the overall framework of AdaptFormer in Figure 2b.

### 3.1 Preliminary and Notation

Vision Transformers (ViTs) are first introduced by [31] into vision recognition. A vanilla vision Transformer basically consists of a patch embedding layer and several consecutively connected encoders, as depicted in Figure 2a. Given an image $x \in \mathbb{R}^{H \times W \times 3}$, the patch embedding layer first splits and flatten the sample $x$ into sequential patches $x_p \in \mathbb{R}^{N \times (P^2 d)}$, where $(H, W)$ represents the *height* and *width* of the input image, $(P, P)$ is the resolution of each image patch, $d$ denotes

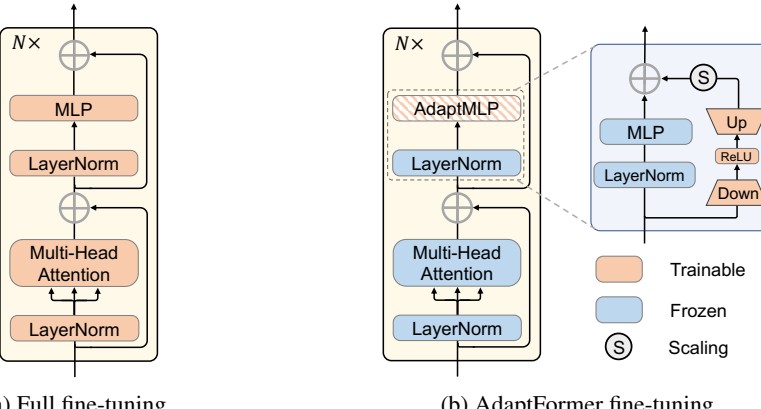

(a) Full fine-tuning.  (b) AdaptFormer fine-tuning

Figure 2: **Comparison of previous *full* and our *AdaptFormer* fine-tuning.** AdaptFormer is conceptually simple by replacing the original MLP block with AdaptMLP, which consists of two branches, including the frozen branch (left) and the trainable down → up bottleneck module (right).

the output channel, and $N = HW/P^2$ is the number of image tokens. The overall combination of a prepended [CLS] token and the image tokens $x_p$ are further fed into Transformer encoders for attention calculation.

Each Transformer encoder mainly consists of two types of sub-layers, *i.e.*, a multi-head self-attention layer (MHSA) and a MLP layer. In MHSA, the tokens are linearly projected and further re-formulated into three vectors, namely $\boldsymbol{Q}, \boldsymbol{K}$ and $\boldsymbol{V}$. The self-attention calculation is performed on $\boldsymbol{Q}, \boldsymbol{K}$ and $\boldsymbol{V}$ by:

$$x'_\ell = \text{Attention}(\boldsymbol{Q}, \boldsymbol{K}, \boldsymbol{V}) = \text{Softmax}(\frac{\boldsymbol{Q}\boldsymbol{K}^\top}{\sqrt{d}})\boldsymbol{V}, \tag{1}$$

where $x'_\ell$ are the tokens produced by MHSA at the $\ell$-th layer. The output tokens $x'_\ell$ are further sent to a LayerNorm [3] and a MLP block which is consisted of two fully connected layers with a GELU activation [45] in between. This process is formally formulated as follows,

$$x_\ell = \text{MLP}(\text{LN}(x'_\ell)) + x'_\ell, \tag{2}$$

where $x_\ell$ is the output of the $\ell$-th encoder block. At the last transformer layer, the [CLS] is utilized for the final object recognition. We refer the readers to find more details in [31]. In our work, we replace the MLP layer with our AdaptMLP module for efficient fine-tuning purposes.

### 3.2 AdaptFormer

We propose a plug-and-play bottleneck module, namely AdaptMLP[2]. We denote the vision Transformer equipped with AdaptMLP as AdaptFormer.

**Architecture.** The design principle of AdaptFormer is simple yet effective, which is illustrated in Figure 2b. Compared to the vanilla full fine-tuning regime, AdaptFormer replaces the MLP block in the transformer encoder with *AdaptMLP*, which is consisted of two sub-branches. The MLP layer in the left branch is identical to the original network, while the right branch is an additionally introduced lightweight module for task-specific fine-tuning. Specifically, the right branch is designed to be a bottleneck structure for limiting the number of parameters purpose, which includes a down-projection layer with parameters $\boldsymbol{W}_{\text{down}} \in \mathbb{R}^{d \times \hat{d}}$, an up-projection layer with parameters $\boldsymbol{W}_{\text{up}} \in \mathbb{R}^{\hat{d} \times d}$, where $\hat{d}$ is the bottleneck middle dimension and satisfies $\hat{d} \ll d$. In addition, there is a ReLU layer [1] between these projection layers for non-linear property. This bottleneck module is connected to the original MLP network (left branch) through the residual connection via a scale factor $s$. For a specific input feature $x'_\ell$, the right branch in AdaptMLP produces the adapted features, $\tilde{x}_\ell$, formally via:

$$\tilde{x}_\ell = \text{ReLU}(\text{LN}(x'_\ell) \cdot \boldsymbol{W}_{\text{down}}) \cdot \boldsymbol{W}_{\text{up}}. \tag{3}$$

---

[2]In this paper, we use the term 'AdaptMLP' to denote the designed module and the term 'AdaptFormer' to represent the fine-tuning framework for Vision Transformers. Unless otherwise specified, we apply AdaptFormer to fine-tune the vanilla ViT backbone [31] in this paper.

Then both the features $\tilde{x}_\ell$ and $x'_\ell$ are fused with $x_\ell$ by residual connection,

$$x_\ell = \text{MLP}(\text{LN}(x'_\ell)) + s \cdot \tilde{x}_\ell + x'_\ell. \tag{4}$$

**Fine-tuning.** During the fine-tuning phase, we only choose the newly added parameters to optimize and keep rest ones fixed. Specifically, the original model parts (blue blocks in Figure 2b) load weights from the pre-trained checkpoint and keeps parameters *frozen*. The newly added parameters (orange blocks) are updated on the specific data domain with the task-specific losses.

**Inference.** After fine-tuning, we still keep the shared parameters frozen as in the previous fine-tuning state, and additionally load the weights of the extra parameters that were fine-tuned in the previous stage. The single overall model is able to be adapted to multiple tasks with the assistance of lightweight introduced modules.

### 3.3  Discussion

**Tunable parameters analysis.** Our AdaptMLP module is lightweight. The total number of parameters introduced to per layer is $2 \times d \times \hat{d} + \hat{d} + d$, which includes biases parameters. The middle dimension $\hat{d}$ is a small value compared with $d$ (AdaptFormer still obtains a decent performance even when $\hat{d} = 1$, as discussed in Sec. 4.5). Since most of the shared parameters are fixed and the number of newly introduced parameters is small ($< 2\%$ of the pre-trained model parameters), the total model size grows slowly when more downstream tasks are added.

**Applicability.** We note that AdaptMLP is a plug-and-play module that can be adaptively inserted into existing popular vision transformer architectures [31, 63, 83, 90, 23, 29] since all of the backbones share the same MLP layers even though they differ in the MHSA architectures (as shown in Figure 2b). Compared to our methods, we notice that recent prompt-related approaches insert trainable parameters into the token space, as illustrated in Figure 3. They prepend learnable parameters either into the embedded tokens before linear projection [58] or the key and value tokens after linear projection [51]. *Therefore, the prompt-related method can not be straightforwardly adapted to special MHSA variants, especially for the one that takes the pyramid spatial information into account* [63, 83]. Besides, we empirically observe that prompt-related methods perform not well when the number of patch tokens grows up from image to video scale, as shown in Figure 1.

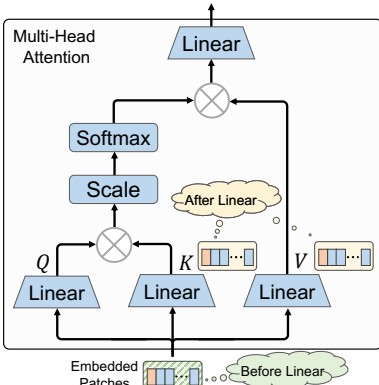

Figure 3: **Prompt tuning illustration.**

In summary, we present a strategy for tuning a pre-trained vision Transformer on a set of scalable vision recognition tasks (*e.g.* image domain and video domain). It adds limited learnable parameters for tuning while achieving comparable or even better performance than the full-tuning strategy. Moreover, AdaptFormer could serve as a generic module for a large variety of recognition tasks.

**Insights of architecture design.** The MLP module is important for ViTs. As illustrated in [30], MLPs prevent ViTs from producing a rank-1 matrix. Also, MLPs stop the ViT output from degenerations. Inspired by the above analysis, we believe an effective ViT adaptation shall focus on its MLPs rather than multi-head self attentions. Meanwhile, we learn from the inception framework [76] that parallel design is an effective way for feature ensemble. With the parallel design, the domain-specific features produced by the adapter module can supplement the domain-agnostic features from the fixed branch for a better feature ensemble. Our following experiments will verify that the parallel performs better than the sequential design.

Besides, though many advanced Transformer-based models [63, 83, 34, 90] which have emerged since the success of ViT having different attention mechanisms within the Transformer block, they all share the similar MLPs (feed-forward network) structures. Therefore, our AdaptMLP can be easily plugged into these ViT variants. Moreover, AdaptMLP can also be applied to more recent attention-free models [77, 61, 19].

# 4 Experiments

We evaluate the effectiveness of AdaptFormer by conducting extensive visual recognition experiments in both the image and video domains. We first describe our experimental settings in Sec. 4.1, covering the pre-trained backbones, baseline methods, downstream tasks and training details. We then compare AdaptFormer with baseline methods and provide a thorough analysis in Sec. 4.2. In addition, we also conduct ablation studies to explore different experimental configurations and explain what makes for the superiority of AdaptFormer in Sec 4.5.

## 4.1 Experimental Settings

**Pre-trained backbone.** We adopt the plain Vision Transformer (ViT) [31], *i.e.*, ViT-Base (ViT-B/16) as our backbone model and pre-train the model with both supervised and self-supervised approaches. Specifically, for **image**, we directly use the ImageNet-21k [26] supervised pre-trained model[3] and MAE [43] self-supervised model[4]. For **video**, we take both supervised and self-supervised pre-trained models from VideoMAE [78]. More details about pre-training approaches and datasets can be found in Appendix.

**Initialization of AdaptFormer.** For the original networks, we directly load the weights pre-trained on the upstream tasks and keep them frozen/untouched during the fine-tuning process. For the newly added modules, the weights of down-projection layers are initialized with Kaiming Normal [44], while the biases of the additional networks and the weights of the up-projection layers are configured with zero initialization. The reason for the zero initialization of other layers is that in this way, the initial newly added parameters are initialized such that the new function resembles the original one at the start of the fine-tuning stage. We empirically found that if the initialization deviates too far from the identity function, the model is not stable to train.

**Baseline methods.** We compare AdaptFormer with three commonly used fine-tuning approaches, including (1)*Linear probing:* adding an extra linear layer on top of the backbone and tuning the added parameters for evaluation. (2) *Full Fine-tuning:* setting all the parameters learnable and tuning them together. (3) *Visual Prompt Tuning (VPT):* [51] fine-tuning the extra token parameters as shown in Figure 3.

**Downstream tasks.** We evaluate our AdaptFormer on both image and video recognition tasks to verify its effectiveness. The specific datasets leveraged in this work are presented in the following.

- **Image domain :** CIFAR-100 [54] contains 50,000 training images and 10,000 validation images of resolution 32×32 with 100 labels. Street View House Numbers (SVHN) [37] is a digit classification benchmark dataset. In total, the dataset comprises over 600,000 labeled images, containing 73,257 training samples, 26,032 testing samples and 531,131 extra training data. The Food-101 [9] dataset consists of 101 food categories with a total of 101k images, including 750 training and 250 testing samples per category.

- **Video domain :** Something-Something V2 (SSv2) [39] is a large collection of video clips showing the people perform several normal actions in the daily life (*e.g.*, moving stuff and opening the door). It consists of 168,913 training samples, 24,777 validation samples and 27,157 testing samples, making a total of 220,847 videos with 174 labels. HMDB51 [55] is composed of 6,849 videos with 51 categories, making a split of 3.5k/1.5k train/val videos.

**Implementation details.** In this work, we use PyTorch toolkit [68] to conduct all experiments on NVIDIA V100 GPUs. Unless otherwise stated, we use 8×8 GPUs for video experiments and 1×8 GPUs for image experiments. Our default configurations follow the *linear probing* settings in [21, 43], which do *not* utilize many common regularization strategies, such as mixup [94], cutmix [91], color jittering and so on. More details can be found in Appendix.

## 4.2 Main Properties and Analysis

We compare the performance of different fine-tuning approaches in Table 1 with the backbones pre-trained via the self-supervised paradigms. The results show that AdaptFormer consistently

---

[3]`https://github.com/rwightman/pytorch-image-models/releases/download/v0.1-vitjx/jx_vit_base_patch16_224_in21k-e5005f0a.pth`

[4]`https://dl.fbaipublicfiles.com/mae/pretrain/mae_pretrain_vit_base.pth`

Table 1: **Fine-tuning with self-supervised pre-trained model.** For tunable parameters, we also report the parameter percentage in the brackets. Besides, we report the top-1 accuracy on different dataset with the absolute value and the gap value relative to the *full-tuning* regime. † denotes $0.1\times$ learning rate due to unstable training.

| Method | Avg. Params (M) | Image | | | Video | |
|---|---|---|---|---|---|---|
| | | CIFAR-100 | SVHN | Food-101 | SSv2 | HMDB51 |
| Full-tuning | 86.04 (100%) | 85.90 | 97.67† | 90.09† | 53.97 | 46.41 |
| Linear | 0.07 (0.08%) | 69.83 (-16.07) | 66.91 (-30.76) | 69.74 (-20.35) | 29.23 (-24.74) | 49.84 (+3.43) |
| VPT [51] | 0.08 (0.09%) | 82.44 (-3.46) | 94.02 (-3.65) | 82.98 (-7.11) | 43.73 (-10.24) | 52.67 (+6.26) |
| AdaptFormer-1 | 0.10 (0.12%) | 83.52 (-2.38) | 93.04 (-4.63) | 83.64 (-6.45) | 50.03 (-3.94) | 51.68 (+5.27) |
| AdaptFormer-4 | 0.15 (0.17%) | 84.83 (-1.07) | 96.19 (-1.48) | 85.42 (-4.67) | 54.70 (+0.73) | 51.81 (+5.40) |
| AdaptFormer-64 | 1.26 (1.46%) | 85.90 (0.00) | 96.89 (-0.78) | 87.61 (-2.48) | 59.02 (+5.05) | 55.69 (+9.28) |

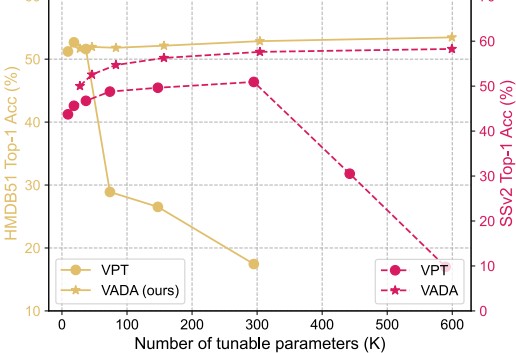 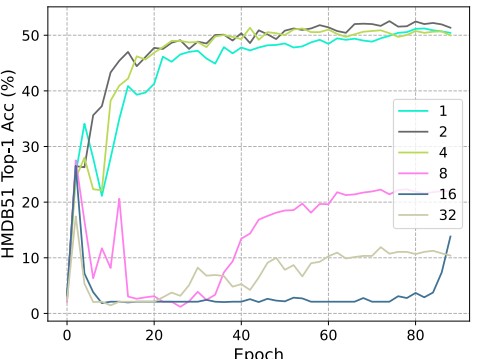

Figure 4: **The trend of performance as the number of tunable parameters grows up.** The accuracy of VPT drops dramatically when the parameter number exceeds task-specific value, while AdaptFormer is robust to the increasing parameters.

Figure 5: **Test accuracy of VPT [51] with different number of introduced tokens.** The optimization procedure becomes unstable when the token number is equal or larger than eight on HMDB51 dataset [55].

surpasses linear probing and Visual Prompt tuning (VPT) methods. Specifically, AdaptFormer-64 outperforms VPT on image benchmark CIFAR-100, SVHN, and Food-101, by 3.46%, 2.87%, and 4.63% respectively. On the more challenging video action recognition dataset Something-Something V2, the superiority becomes even more significant, *i.e.*, about 15%. Note that even compared with the full fine-tuning strategy, our AdaptFormer still outperforms by about 5% Top-1 accuracy on SSv2 dataset. To summarize, our AdaptFormer is highly parameter-efficient, as well as yielding good performance with parameter size at most 2% times than the full fine-tuning manner.

## 4.3 Scaling Tunable Parameters Up

Even though there are only limited parameters introduced, one might also argue that more tunable parameters of AdaptFormer contribute to its higher accuracy compared with VPT [51]. We conduct experiments to make a comprehensive discussion on this aspect.

As described in Sec. 3.3, the number of tunable parameters can be adjusted by changing the number of introduced tokens for VPT, or the hidden feature dimension for AdaptFormer. As shown in Figure 4, we conduct experiments with a wide range of tunable parameters on both SSv2 and HMDB-51 datasets. Since AdaptFormer and VPT share the same number of parameters of classification head on a specific dataset, we only report the tunable parameters on the x-axis, which comes from the visual prompts (VPT) or weight/bias of the down-up fully-connected layers (AdaptFormer), without calculating the parameters of classification head. For VPT, the number of introduced tokens is chosen from {1, 2, 4, 8, 16, 32, 48, 64}. Similarly, the number of hidden dimensions in AdaptFormer is in {1, 2, 4, 8, 16, 32}. AdaptFormer has a slight performance gain or maintains the accuracy stably when the parameters scale up. On the contrary, the performance of VPT decreases dramatically when the parameters exceed the task-specific value. Moreover, choosing the most suitable number of token

Table 2: **AdaptFormer for multi-label classification.**

| Method | Params (M) | NUS-WIDE [24] |
|---|---|---|
| Full-tuning | 85.86 (100%) | 61.26 |
| Linear | 0.06 (0.08%) | 51.19 (-27.25) |
| VPT [51] | 0.07 (0.09%) | 57.08 (-7.56) |
| AdaptFormer-1 | 0.09 (0.12%) | 57.51 (-4.08) |
| AdaptFormer-4 | 0.15 (0.17%) | 58.14 (-2.13) |
| AdaptFormer-64 | 1.25 (1.46%) | 59.07 (-0.06) |

number becomes laborious since it might be task-specific (*i.e.* varying from one dataset to the other one). For example, the accuracy of VPT keeps going up when the number of tunable parameters increases up to 300K on SSv2, whereas it begins to drop when the number of tunable parameters exceeds 50K on HMDB-51.

We further study the optimization procedures of VPT by monitoring the test accuracy of the training stage. As shown in Figure 5, we gradually increase the number of tokens in VPT and plot the Top-1 accuracy of each epoch. The training stages are stable when the number of tokens is less than or equal to 4, *e.g.*, {1, 2, 4}. However, when the number becomes 8 or larger, *e.g.*, {8, 16, 32}, the training procedure collapses at about the tenth epoch and achieves poor performance at the end of the training stage. On the contrary, the optimization procedures of AdaptFormer are stable when the number of parameters varies across a large range, as shown in Table 3a. The top-1 accuracy fluctuates within 1.5% when the number of parameters increases from 0.44M (`dim=16`) to 4.87M (`dim=256`).

## 4.4 Multi-Label Classification

We further conduct experiments on dataset with larger scale and diversity. Specifically, we evaluate AdaptFormer on NUS-WIDE [24] for multi-label classification. NUS-WIDE contains 269,648 images collected from Flicker, which are annotated with 81 visual concepts. Since some images are not available on Flicker, we only use 220,000 images following [7, 32]. We utilize mean average precision (mAP) as performance metric.

**Settings and results.** Our training settings mainly follow ASL [7]. Specifically, We trained all models for 40 epochs using Adam optimize and 1-cycle learning rate policy [73]. The maximal learning rate is 0.001. As shown in Table 2, though AdaptFormer-64 achieves a slightly lower mAP than fine-tuning, it significantly reduces the amount parameters that need to be updated (from 85.86 to 1.25M). Moreover, AdaptFormer has an clear advantage over other fine-tuning approaches including linear probing and VPT.

## 4.5 Ablation Studies

We ablate our AdaptFormer to study what properties make for a good AdaptFormer and observe several intriguing properties. The ablation studies conducted in this work are all performed on the SSv2 validation set [39].

Table 3: **AdaptFormer ablation experiments** with ViT-B/16 on SSv2. We report the top-1 accuracy on the `val` set. Most suitable settings are marked in color.

(a) **Middle dimension** $\hat{d}$.

| mid dim | #params | top-1 |
|---|---|---|
| 1 | 0.16M | 50.03 |
| 16 | 0.44M | 57.62 |
| 32 | 0.73M | 58.27 |
| 64 | 1.32M | **59.02** |
| 256 | 4.87M | 58.87 |

(b) **AdaptMLP inserted layers and form.**

| layers | form | #params | top-1 |
|---|---|---|---|
| $1 \rightarrow 6$ | parallel | 0.73 | 50.48 |
| $7 \rightarrow 12$ | parallel | 0.73 | 57.99 |
| $1 \rightarrow 12$ | parallel | 1.32 | 59.02 |
| $1 \rightarrow 12$ | sequential | 1.32 | 58.17 |

(c) **Scaling factor** $s$.

| factor | top-1 |
|---|---|
| 0.01 | 53.44 |
| 0.05 | 58.85 |
| 0.10 | 59.02 |
| 0.20 | 58.89 |

**Middle dimension.** The middle dimension controls the number of introduced parameters by Adapt-Former. Lower middle dimensions introduce fewer parameters with a possible performance cost. We ablate AdaptFormer on the middle feature dimension to study this effects. As shown in Table 3a, the accuracy consistently improves when the middle dimension increases up to 64 and reaches the saturation point when the middle dimension is about 64 on SSv2 dataset. We note that our AdaptFormer can achieve a decent performance when the middle dimension reduces even to one, about 50.03% top-1 accuracy.

We conduct more extensive ablation studies on middle dimension in Appendix Table 10 and found that the optimal middle dimension varies per dataset. For example, the accuracy reaches saturation when the middle dimension equals 64 on SSv2, whereas for NUS-WIDE dataset, the mAP slightly improves when the middle dimension increases from 64 to 512. However, AdaptFormer with middle dimension as 512 has 0.75 mAP higher (59.82 vs. 59.07 mAP) than the one with 64 at the cost of about 8 times more parameters. Therefore, we choose the `middle dimension=64` for both SSv2 and NUS-WIDE for a better trade-off.

**Scaling factor.** The scaling factor $s$ is introduced to balance the *task-agnostic* features (generated by the original frozen branch) and the *task-specific* features (generated by the tunable bottleneck branch). We evaluate AdaptFormer with multiple $s$ values and the results are summarized in Table 3c. Different from the scaling factor in NLP field which prefer $s$ larger than 1 (*e.g.*, $s = 4$ in [42]), we empirically found that the $s$ should be $< 1$ for vision tasks, otherwise the fine-tuning would become unstable. Besides, we found that AdaptFormer achieves optimal performance with $s = 0.1$. A larger or smaller $s$ would bring slight performance drop. Thus, we choose $s = 0.10$ as a default setting.

**AdaptFormer position.** As shown in Table 3b, we further ablate on the specific position to introduce the AdaptMLP block. We gradually increase the number of AdaptMLP layers with a step of three (start → end, both included). We observe that the performance of AdaptFormer has a positive correlation with the number of added layers. In addition, AdaptFormer prefers the top part (the one far away from the input image) of the network to the bottom part when introducing the same number of layers, *e.g.*, AdaptFormer with 7 → 12 obtains over 14.5% higher accuracy than 1 → 6, though both equipped with six AdaptMLP layers.

**Insertion form.** We study the insertion formulation by comparing the *parallel* and *sequential* instances which are illustrated in Figure 6. As shown in Table 3b, the parallel AdaptFormer is able to outperform the sequential one by 0.85% top-1 accuracy. The reason might be: **(1)** the parallel design maintains the original feature using an independent branch and aggregating updated context by element-wise scaled sum; **(2)** the sequential design is equivalent to adding more layers, which might cause optimization difficulty. Therefore, we adopt the parallel design as our default setting due to its superiority.

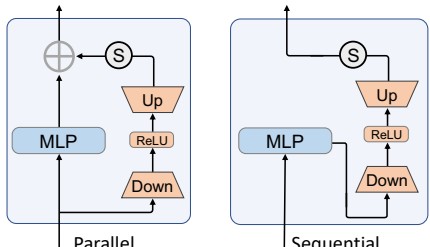

Figure 6: **Illustration of the *parallel* and *sequential* insertion form**. Comparison results are shown in Table 3b.

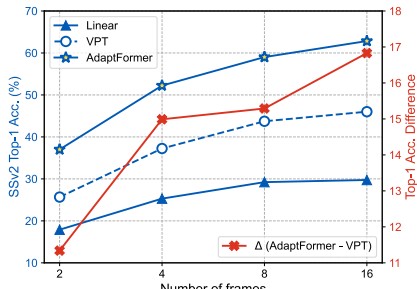

Figure 7: **Performance with video frames number.** AdaptFormer outperforms VPT and linear fine-tuning.

**Number of frames.** The number of embedded patch tokens increases linearly with the number of video frames for the plain ViT [31]. We conduct experiments with the different number of frames, *i.e.*, {2, 4, 8} and the results are shown in Figure 7. We observe that increasing the number of frames is beneficial for all these three fine-tuning methods. However, AdaptFormer consistently outperforms the linear manner (*e.g.*, +30% top-1 accuracy on 8 input frames) and VPT method(*e.g.*, +14% top-1 accuracy on 8 input frames).

### 4.6 Towards Visual Recognition Generalist Agent

In the above experiments, we typically utilize a modality-specific pre-trained checkpoint for the corresponding downstream tasks. For example, we use Kinetics-400 (**video domain**) pre-trained model for downstream video action recognition on Something-Something V2 and HMDB-51 benchmarks. Besides, we use ImageNet-21K (**image domain**) pre-rained model for downstream image classification on CIFAR-100, SVHN and Food-101 benchmarks. Our AdaptFormer achieves superior performances in this *same network with modality-specific weights* scenario.

Next, we take a further step to ask what would happen if using *the same network with the modality-agnostic weights* for multiple tasks in the multi-modalities downstream tasks?

We use the model pre-trained on ImagNet-21k to do action recognition on SSv2. As shown in Table 4, AdaptFormer is robust to domain shift caused by modality. The experimental results show that the linear probe approach obtains a very poor accuracy (*i.e.*, 6.56% top-1 accuracy) when fine-tuning on SSv2. Meanwhile,

Table 4: **Fine-tuning on video data with image** pre-trained model.

| Method | Avg. Params (M) | Fine-tuning SSv2 |
|---|---|---|
| Full-tuning | 86.36 | 41.50 |
| Linear | 0.15 | 6.56 |
| VPT [51] | 0.16 | 16.94 |
| AdaptFormer | 1.33 | 46.06 |

VPT [51] achieves a better performance than linear probe but it is not decent (*i.e.*, 16.94% top-1 accuracy). Our AdaptFormer, compared to the above two methods, attains a promising 46.06% top-1 accuracy, which is even higher than the full-tuning schedule (+4.56%).

### 4.7 Visualization

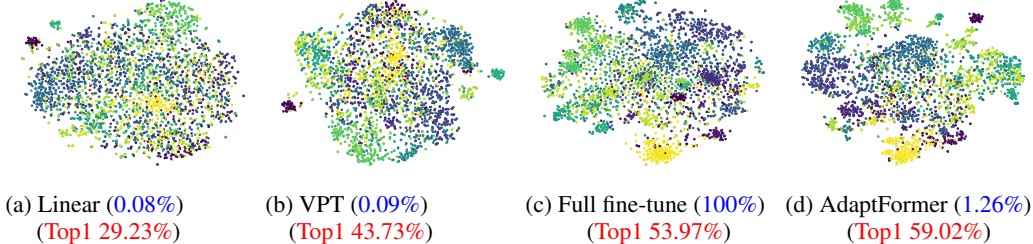

(a) Linear (0.08%)
(Top1 29.23%)

(b) VPT (0.09%)
(Top1 43.73%)

(c) Full fine-tune (100%)
(Top1 53.97%)

(d) AdaptFormer (1.26%)
(Top1 59.02%)

Figure 8: **t-SNE visualizations on SSv2 val dataset.** We extract the final classification features from the top linear layer for t-SNE visualizations. The top-1 accuracy is reported in red, while the relative parameter (compared to the full fine-tuning strategy) is reported in blue.

To evaluate the quality of the produced features, we conduct t-SNE [80] visualizations on Adapt-Former and other baseline methods. The features are extracted from the SSv2 validation set via the ViT-Base backbone. Figure 8 shows that the linear fine-tuning and the VPT methods tend to output mixed features as shown in Figure 8(a)-(b). Compared with the above two methods, the full fine-tuning strategy performs well in projecting features. However, it consumes huge computational sources to tune the whole network parameters. Figure 8(d) validates that our AdaptFormer facilitates ViT-Base in generating more separable representations with fewer learnable parameters.

## 5 Conclusion

We present a conceptually simple yet effective framework, AdaptFormer, for efficiently adapting a pre-trained Vision Transformer (ViT) backbone to scalable vision recognition tasks. By introducing AdaptMLP, our AdaptFormer is able to fine-tune the lightweight modules for producing features adapted to multiple downstream tasks. The extensive experiments on five datasets, covering both the image and the video domains, validate that our proposed methods are able to increase the ViT's transferability with little computational cost. We hope our work will inspire future research in exploring more efficient fine-tuning methods for large vision models. One limitation is that AdaptFormer is only employed in recognition tasks in this work, it's unclear whether it can work well in tasks beyond recognition, *e.g.*, object detection and semantic segmentation. We leave it for the future exploration. Since our method is specially designed for efficient fine-tuning, we do not foresee obvious undesirable ethical/social impacts at this moment.

**Acknowledgment.** This work is supported by CCF-Tencent Open Fund. Ping Luo is supported by the General Research Fund of HK No.27208720, No.17212120, and No.17200622.

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
