# OpenReview forum: "AdaptFormer: Adapting Vision Transformers for Scalable Visual Recognition"
_NeurIPS.cc/2022/Conference — NeurIPS 2022 Accept_

### Official Review · Reviewer_KxCH · 2022-07-10

**Rating:** 4
**Confidence:** 4
**Soundness:** 2 fair
**Presentation:** 3 good
**Contribution:** 3 good

**Summary:**

The paper introduces an effective adaptation approach for vision Transformer which avoids the training cost of large-scale parameters in downstream task fine-tuning. In detail, the AdaptMLP is devised by including an encoder-decoder branch into the original MLP layer in the self-attention block. During the fine-tuning of the downstream task, only the parameters of the AdaptMLP module are optimized and all other parameters are frozen. Experiments of the inner-domain (image/video) pretraining task with five different downstream datasets and the cross-domain pretraining task between ImageNet-21k and SSv2 datasets, validate the proposed approach.

**Questions:**

1.	What is the rationale behind the design of the AdaptMLP module? Dose the structure of encoder-decoder perform the best? How about other structures for the design of this module? Is there any empirical reason or theory support such design?
2.	How about the generalization ability of the AdaptFormer on large-scale image datasets, e.g., NUS-WIDE?
3.	Why the Transformer pre-trained on ImageNet-21k performs such poor (41.5%) on Something-Something V2 datasets? In action recognition research filed, the model pre-learned on image datasets (i.e., ImageNet) will even perform better than the model pre-learned on video dataset (e.g., Kinetics). The performance is questionable.


**Limitations:**

The major limitation of this work is that the rationale behind the AdaptMLP module is unclear. There is no theory or reference mentioned in the paper that validates the reasonability of such module design. Only listing the performance gains is not enough. The evaluation is not very convincing since the downstream image datasets are very small and not very diverse.

**Strengths And Weaknesses:**

[Strengths]

1.	The paper forms well and the technical implementation details are clear.
2.	It is a good direction to reuse the parameters of the vision Transformer pre-trained on the ImageNet for the downstream tasks. The problem is very close to the real-world application and much valuable in the research community.
3.	The performances on the image and video classification tasks give some clues that the proposed AdaptMLP module can learn some task-specific knowledge to enhance the feature learning.

[Weakness]

1.	The biggest issue for this work is about the design of AdaptMLP. The rationale behind the newly-proposed encoder-decoder branch is unclear. The technical part from 152-165 only shows the implementation details and it is very hard to judge the technical contributions for this module.
2.	The evaluated downstream image datasets (e.g., CIFAR-100 and SVHN) are very small and the relative image contents are not diverse. The classification performances on these two datasets are saturated. Thus, the evaluations are not very convincing in my opinion.
3.	In the experimental section, there should be more analysis rather than only describing the results. For instance, in the investigation of the middle dimension in the encoder-decoder block, there is no point that gives some insights to readers how reasonable the choice it is. The dimension of 64 is the best and I don’t know why.
4.	The t-SNE visualization in Figure 8 cannot verify the effectiveness of AdaptFormer compared to the full-finetune since the (c) and (a) are almost the same.

---

> ### Author Response · Authors · 2022-08-02
> **Response to Reviewer KxCH (Part 1/2)**
>
>
> Dear Reviewer KxCH,
>
> Thank you for your valuable comments. We answer the raised questions below.  We have included all discussions and results in our revised manuscript (blue content).
>
>
> **1. The rationale for the design of adapter module**
>
> We illustrate the rationale for our adapter design from two perspectives. First, we illustrate our motivation for designing a **parallel** branch to the original MLP module. Second, we illustrate our motivation for designing an **encoder-decoder MLP** structure for our AdaptFormer.
>
>
> - The MLP module is important for ViTs. As illustrated in [a], MLPs prevent ViTs from producing a rank-1 matrix. Also, MLPs stop the ViT output from degenerations. Inspired by the above analysis, we believe an effective ViT adaptation shall focus on its MLPs rather than multi-head self attentions. Meanwhile, we learn from the inception framework [b] that parallel design is an effective way for feature ensemble. With the parallel design, the domain-specific features produced by the adapter module can supplement the domain-agnostic features from the fixed branch for a better feature ensemble. Our experiments verified that the parallel performs better than the sequential design (Sec.4.4 **Insertion form**).
>
> - Our Adapter is built up with the encoder-decoder structure of MLPs. We choose MLPs since they are the most basic architectures in deep neural network designs [c]. We verified the effectiveness of MLPs by comparing MLPs with some other structures (Appendix **A.4 Possible Architecture** and below table). The lightweight *encoder-decoder* structure aims to limit newly introduced parameters by reducing the middle dimension to a small value (`middle dimension=64` in our default setting).
>
> - Though having different attention mechanisms within the Transformer block, ViT and its variants (e.g., Swin, PVT) all share the same MLPs (feed-forward network) structures. Therefore, our AdaptMLP can be easily plugged into these ViT variants.
>
> Besides rationale analysis, we provide an evaluation on the design choice of adapter structure below. We design some extensions of our AdaptFormer by taking the spatial and channel-wise properties of visual inputs. Specifically,
>
> - **(1.)** We insert a 2D convolution layer with `kernel size=3` and `stride=1` after the non-linear function of our introduced bottleneck structure;
> - **(2.)** Similar to **(1.)**, we replace the vanilla 2D convolution with a depthwise convolution layer, reducing the number of parameters and FLOPs of the convolution layer;
> - **(3.)** We add a LayerNorm to the input of the newly introduced AdaptMLP branch;
>
> We comprehensively compare these structure variants on multiple tasks, including SSv2 video classification, NUS-WIDE image multi-label classification, and CIFAR100 image classification.
>
> | method         | Avg Params | SSv2 Top1 | NUS-WIDE mAP | CIFAR100 Top1 |
> | -------------- | ---------- | --------- | ------------ | ------------- |
> | Conv           | 1.39       | 58.47     | 58.8573      | 85.42         |
> | Depthwise Conv | 1.29       | 58.15     | 58.7262      | 85.37         |
> | LayerNorm-In   | 1.30       | 57.85     | 58.5123      | 85.71         |
> | AdaptFormer    | 1.28       | 59.02     | 59.0663      | 85.93         |
>
>
> **Discussion:** As shown in the above table, our AdaptFormer consistently outperforms other variants on these three benchmarks. Therefore, we recommend our default simple yet effective AdaptFormer design.
>
>
> We have included the above analysis and results in  Appendix **A.4 Possible Architectures** in the revised manuscript.
>
>
> **2. Evaluation on the datasets with larger scale and diversity**
>
> Thanks for your suggestion. To further evaluate AdaptFormer on larger-scale and more diverse datasets, we conducted experiments on NUS-WIDE [f] for multi-label classification. The results are shown in the following table, which validates that our proposed method is able to achieve comparable results to full-tuning method with only 1.25M tunable parameters. Besides, our AdaptFormer outperforms linear probing and VPT by a certain margin. We have included this result in Appendix **A.3 Multi-Label Classification** in the revised manuscript.
>
> | Method     | #Params | mAP     |
> | ---------- | ------- | ------- |
> | Full       | 85.86   | 61.2597 |
> | Linear     | 0.06    | 51.1908 |
> | VPT        | 0.07    | 57.0853 |
> | Adapter-1  | 0.09    | 57.5072 |
> | Adapter-4  | 0.15    | 58.1446 |
> | Adapter-64 | 1.25    | 59.0663 |
>
> Furthermore, in addition to image tasks, we extend our proposed framework to video benchmarks (SSv2 and HMDB) in our original paper (Table.1 in the main paper).

---

> > ### Author Response · Authors · 2022-08-02
> > **Response to Reviewer KxCH (Part 2/2)**
> >
> >
> > **3. Insights for the design of the middle dimension**
> >
> > We thank you for the *more analysis* suggestion. For the middle dimension design, we aim to seek a trade-off between model capacity (i.e., potential) and adaptation efficiency. The middle dimension has the main influence on the parameter size of the adapter. The higher dimension brings more parameters while the efficiency and storage are limited. We evaluate several numbers of middle dimension and found that using 64 is optimal to achieve accuracy, light-weight storage, and efficiency. We have included these results in Appendix **A.6 Extended experiments on middle dimension** in the revised manuscript.
> >
> >
> >
> >
> > | Middle Dim  | Params  | SSv2 Top1  | NUS-WIDE mAP  |
> > |-------------|---------|------------|---------------|
> > | 1           | 0.16    | 50.03      | 57.5072       |
> > | 4           | 0.22    | 54.70      | 58.1446       |
> > | 16          | 0.44    | 57.62      | 59.0003       |
> > | 32          | 0.73    | 58.27      | 59.0873       |
> > | 64          | 1.32    | 59.02      | 59.0663       |
> > | 128         | 2.51    | 58.95      | 59.4929       |
> > | 256         | 4.87    | 58.87      | 59.6230       |
> > | 512         | 9.59    | 58.98      | 59.8214       |
> >
> >
> >
> > **4. Clarifications on t-SNE visualization.**
> >
> > Perhaps a misunderstanding. We agree that the visualization of (c) and (a) are similar since the classification performance of (c) and (a) is similar. However, our AdaptFormer takes great advantage in the network parameters. Specifically, there are only 1.26M parameters needed to be fine-tuned in AdaptFormer, covering only 1.46% parameters in the full fine-tuning strategy.
> > We have added the parameters comparisons to Figure 8 for a clear illustration.
> >
> >
> >
> > **5. Clarification of performance on Something-Something V2 datasets.**
> >
> > Perhaps a misunderstanding. The comparison paradigms and training configurations are different between the mentioned works and our AdaptFormer. Specifically, previous works utilize many common regularization strategies (e.g. mixup, cutmix, drop path and color jittering) to achieve better performance. However, these strong regularization strategies are harmful to linear probing experiments.  In our work, for apple-to-apple comparisons, we completely follow the linear probing training settings in [d] and [e] to conduct the experiments in images and videos, respectively, which disable above mentioned regularization strategies following [g].
> >
> >
> > ---
> > ### Reference
> >
> > [a] Dong, Yihe, et al. "Attention is not all you need: Pure attention loses rank doubly exponentially with depth." ICML 2021.
> >
> > [b] Szegedy, Christian, et al. "Going deeper with convolutions." CVPR 2015.
> >
> >
> > [c] Hornik, Kurt, Maxwell Stinchcombe, and Halbert White. "Multilayer feedforward networks are universal approximators." Neural networks 1989.
> >
> > [d] He, Kaiming, et al. "Masked autoencoders are scalable vision learners." CVPR 2022.
> >
> > [e] Tong, Zhan, et al. "Videomae: Masked autoencoders are data-efficient learners for self-supervised video pre-training." arXiv 2022.
> >
> > [f] Chua, Tat-Seng, et al. "Nus-wide: a real-world web image database from national university of singapore." Proceedings of the ACM international conference on image and video retrieval. 2009.
> >
> > [g] Chen, Xinlei, et al. "An empirical study of training self-supervised vision transformers." ICCV 2021.

---

> > > ### Comment · Reviewer_KxCH · 2022-08-08
> > > **Thanks for your reply**
> > >
> > > Thanks for the detailed responses to my questions. In the aspect of the rationale behind the architecture design, I believe that the MLPs is the key point to address the issue of parameter adaptation. Nevertheless, such design is seemingly a heuristic engineering work without in-depth insights. In the part of experiments, the settings of exploring different module structures are interesting and the results are reasonable. The evaluation on NUS-WIDE is also fine. One remained issue is about the explanation on the performance of action recognition. Employing augmentation indeed improves the final performance. However, even under the same video augmentation, the performance of the model pre-trained on ImageNet can still outperform the model pre-trained on Kinetics in my experience. With these in mind, I would like to increase my score to 4.

---

> > > > ### Author Response · Authors · 2022-08-09
> > > > **Response to Reviewer KxCH (Part 1/2)**
> > > >
> > > >
> > > >
> > > > Dear Reviewer KxCH,
> > > >
> > > >
> > > > Thank you very much for your feedback. We are glad to see that our response has addressed your concerns about the generalization ability of the AdaptFormer on large-scale image datasets and exploration of other structures. We answer the further raised questions below.
> > > >
> > > >
> > > > **Q1. In-depth insights of architecture design**
> > > >
> > > > A1: We are glad to see that you agree that MLPs are the key point to address the issue of parameter adaptation. We would like to elucidate further our claim (i.e., *We choose MLPs since they are the most basic architectures in deep neural network designs*) to address your concerns about in-depth insights.
> > > >
> > > > - This claim is indeed inspired by [ii] where the abstract says:
> > > >
> > > > > This paper rigorously establishes that standard multilayer feedforward networks with as few as one hidden layer using arbitary squashing functions are capable of approximating any Borel measurable function from one finite dimension space to another any desired degree of accuracy, provided sufficiently many hidden units are available. In this sense, multilayer feedforward networks are a class of universal approximators.
> > > >
> > > > This abstract has shown that MLPs with nonlinear activation functions are able to approximate universal transformations from one feature space to another, which is beneficial to the general feature transformation tasks.
> > > >
> > > > - Besides, the MLPs in the original ViT model are connected among the channel dimensions. Accordingly, we design a parallel branch where MLPs are set with similar channel connections to effectively transfer the MLPs input from the current dataset domain to the target dataset domain. The effective transformations convince us to choose MLPs as our adapter core architecture. Moreover, as we aim to design a lightweight adapter module with as few parameters as possible for efficient transfer and lightweight storage, we choose the encoder-decoder architecture of MLPs for parameter reduction purpose.
> > > >
> > > > - Moreover, we adopt the parallel design instead of the sequential design since we learn from [iv] that the *“parallel structures mitigate the impact of structural changes on nearby components”*. That's to say, the parallel design facilitates the domain-specific features (from the adapter module) in supplementing domain-agnostic features (from the nearby backbone) for better feature transformation. In contrast, the sequential designs may disturb the original domain-agnostic features and have a catastrophic impact on feature transformation.

---

> > > > > ### Author Response · Authors · 2022-08-09
> > > > > **Response to Reviewer KxCH (Part 2/2)**
> > > > >
> > > > >
> > > > > **Q2. Clarification of the performance difference with ImageNet and Kinetics pre-trained weights**
> > > > >
> > > > > A2. The type of spatiotemporal attention (**divided** *vs.* **joint**) determines whether the performance of the model pre-trained on ImageNet can outperform the model pre-trained on Kinetics.
> > > > >
> > > > > - A similar phenomenon has been discussed in recent work, Uniformer [i], independently. We borrow the experimental results from Table 4(c) in Uniformer paper [i] and summarize them in below **Table 1**. Specifically, the **divided** spatiotemporal attention prefers ImageNet to Kinetics-400 for the pre-training dataset. The performance of the **divided** attention model pre-trained on ImageNet outperforms the model pre-trained on Kinetics-400. On the contrary, the **joint** spatiotemporal attention prefers Kinetics-400 to ImgaeNet. The **joint** attention model attains higher top-1 accuracy with Kinetics-400 pretraining compared to ImageNet (53.8 *vs.* 52.0).
> > > > >
> > > > > - We adopt the **joint** spatiotemporal attention for all video-related experiments in this work (introduced in **Appendix A.1.1**). Therefore, our experimental phenomenon is consistent with the joint attention in [i], i.e., Kinetics pretraining is preferable.
> > > > >
> > > > >
> > > > >
> > > > > **Table 1:** Comparisons of different spatiotemporal attention types with ImageNet and Kinetics pre-trained weights. **Note:** Divided Attention means that temporal attention and spatial attention are separately applied one after the other. On the contrary, Joint attention computes the spatial and temporal relationships simultaneously. More details can be found in TimeSformer [iii].
> > > > >
> > > > > | Spatiotemporal Attention | Pretrain | SSv1 Top-1 | Top-1 Difference (K400-ImageNet) |
> > > > > | ------------------------ | -------- | ---------- | ---------------- |
> > > > > | Divided                  | ImageNet | 51.9       | -                |
> > > > > | Divided                  | K400     | 51.8       | **-0.1**         |
> > > > > |                          |          |            |                  |
> > > > > | Joint                    | ImageNet | 52.0       | -                |
> > > > > | Joint                    | K400     | 53.8       | **+1.8**         |
> > > > >
> > > > > We have added these discussions into the updated manuscript (blue content in Appendix **A.8**).
> > > > >
> > > > > We hope our response has resolved your concerns and convinced you of the merits of our work. **And we sincerely appreciate that if you could consider increasing the rating accordingly.**
> > > > >
> > > > > We are always willing to address any of your further concerns.
> > > > >
> > > > >
> > > > > Thank you for your time.
> > > > >
> > > > > Best wishes,
> > > > >
> > > > > Authors
> > > > >
> > > > > ---
> > > > >
> > > > > ### Reference
> > > > >
> > > > > [i] Li, Kunchang, et al. "Uniformer: Unified transformer for efficient spatiotemporal representation learning." ICLR 2022.
> > > > >
> > > > > [ii] Hornik, Kurt, Maxwell Stinchcombe, and Halbert White. "Multilayer feedforward networks are universal approximators." Neural networks 1989.
> > > > >
> > > > > [iii] Bertasius, Gedas, Heng Wang, and Lorenzo Torresani. "Is space-time attention all you need for video understanding?." ICML 2021.
> > > > >
> > > > > [iv]. Christian, Szegedy et al. "Rethinking the Inception Architecture for Computer Vision." CVPR 2016.

---

> ### Author Response · Authors · 2022-08-08
> **Sincerely Look Forward to Your Feedback**
>
> Dear Reviewer KxCH,
>
>
> Thanks again for all of your constructive suggestions, which have helped us improve the quality and clarity of our paper.
> We hope that the new experiments and additional explanations have convinced you of the merits of our work, and turns your assessment to the positive side.
>
> As the deadline for discussion is approaching, please do not hesitate to contact us if there are other clarifications or experiments we can offer.
>
> Best regards,
>
> Authors

---

### Official Review · Reviewer_9rzH · 2022-07-10

**Rating:** 5
**Confidence:** 4
**Soundness:** 2 fair
**Presentation:** 3 good
**Contribution:** 3 good

**Summary:**

The paper propose a simple method to adapt a pre-trained model to down-streaming tasks with most of the parameters frozen. The authors propose to use a SE attention like extra branch to adjust the MLP block when fine-tuned on downstream tasks. Overall, the paper is well written and the experiments are comprehensive. However, some of the experiment settings are not clear at the moment.

**Questions:**

1. Based on the results shown in table 1, the ViT-B model only achieves 85.99% accuracy on CIFAR100. However, in other papers such as DeiT, a pre-trained ViT model can easily achieve 90%+ accuracy.

2. Why the proposed methods achieve much better accuracy than the full-tuning baseline with much less number of tunable parameters?

3. The main motivation of the paper is the huge memory and computation cost of the traditional full-tuning method. How much of memory and computations could the proposed method save on those aspects?

4. Since the authors are using the ImageNet-21K pretrained weights, why no experiments are conducted on ImageNet-1k dataset?

**Limitations:**

NA.

**Strengths And Weaknesses:**

Strength:

1. the paper is well motivated. Adaptation of large pre-trained ViT models is an important topic to make the recent popular ViT models more useful.
2. the paper is well written and easy to follow. The presentation is clear and the proposed method is well illustrated.
3. the experiments results are promising. The proposed method achieves significantly better results over the full finetuning baselines.

Weakness:

1. the settings of the full fine-tuning are not clear. The results of AdaptFormer uses pre-trained weights on ImageNet-21k. Does the full-tuning baseline also uses the pre-trained weights on large dataset?

2. I was not clear on why the proposed method could achieve better results with most of the backbone parameters frozen. With more parameters tunable, the full-tuning should have higher representation capability. This counter intuitive results are not well explained in the paper.

---

> ### Author Response · Authors · 2022-08-02
> **Response to Reviewer 9rzH (Part 1/2)**
>
>
> Dear Reviewer 9rzH,
>
> Thank you for your valuable comments. We answer the raised questions below. We also include our revision (blue content) in the paper.
>
>
> **1.Detailed settings of the full fine-tuning**
>
> Thanks for the comments. The results of AdaptFormer and full-tuning utilize the same pre-trained weights in the same column of a specific table for fair comparisons. However, we compare AdaptFormer with full-tuning under various scenarios with not only ImageNet-21 pretrained weight but also ImageNet-1K, Kinetics-400. We utilize both self-supervised and supervised pre-trained models. The corresponding pretraining datasets for image and video are summarized below.
>
> | -     | Supervised Pretrain | Self-Supervised Pretrain |
> | ----- | ------------------- | ------------------------ |
> | Image | ImageNet-21K        | ImageNet-1K              |
> | Video | Kinetics-400        | Kinetics-400             |
>
> Besides, the training settings of AdaptFormer and full-tuning are strictly the same for fair comparisons. We document these detailed settings in Appendix A.1.
>
>
>
>
> **2.Intuitive reasons behind the results**
>
> Thanks for the valuable comments. Theoretically speaking, with more tunable parameters, the networks should own more capacities to improve the classification performance. However, in practice, it will be harder for the networks trained with more parameters to be well optimized. Unlike the full fine-tuning strategies, our AdaptFormer is more efficient to be optimized, thus achieving better performance. Besides, the lightweight tunable modules make our AdaptFormer more practical in real-world applications.
>
> **3.Clarifications on different performances between AdaptFormer and DeiT**
>
> The pre-trained weights utilized in AdaptFormer and DeiT are different. Specifically, DeiT load the supervised pre-trained weights on ImageNet-1K, while AdaptFormer utilized the MAE pre-trained weights instead. In Table 2 (Appendix), when using ImageNet 21K pre-trained model, we get 91.86% on CIFAR100.  It’s empirically found that MAE pre-trained model brings lower fine-tuning accuracy on CIFAR100.
>
> **4. Memory and computation that AdaptFormer can save compared with Full-tuning**
>
> Thanks for your suggestion. We study the empirical computation cost and memory usage at both fine-tuning and inference stages. All experiments in this part use the same NVIDIA A100-40G GPU. We evaluate AdaptFormer on SSv2 video classification with a `batch size=32` and `num frames=4`. All the time is measured in milliseconds averaged over 100 trials.  The results are summarized in Table 1 and Table 2.
>
> **Table 1** Finetuing stage.
>
> | method      | Latency (ms)   | Memory (GB)   |
> | ----------- | -------------- | ------------- |
> | Full-tuning | 355.0          | 38.7          |
> | AdaptFormer | 251.7 (-29.1%) | 24.9 (-35.7%) |
>
>
> **Table 2** Inference stage.
>
> | method      | Latency (ms) | Memory (GB)   |
> | ----------- | ------------ | ------------- |
> | Full-tuning | 42.3         | 4.20          |
> | AdaptFormer | 42.8 (+1.2%) | 4.22 (+0.48%) |
>
>
> **Discussion:** As shown in Table 1, since most pre-trained parameters are frozen, AdaptFormer significantly reduces the training time (-29.1%) and GPU memory usage (-35.7%) compared with the full-tuning. At the inference stage, the newly added branch of AdaptFormer will introduce slight computation cost and GPU memory usage. However, since the AdaptMLP module is designed to be lightweight, this additional burden is negligible.
>
> We note that the most significant benefit of AdaptFormer is the reduction in storage usage. Taking ViT-Base (~86M) for example, if there are 100 tasks, the corresponding AdaptFormer needs `(86+1.3x100) x 10^6 x 4(bytes) x (1/10^-6)(MB/bytes) = 864 (MB)` storage memory. However, full-tuning 100 models independently requires `86 x 100 x 10^6 x 4(bytes) x (1/10^-6)(MB/bytes) = 34400(MB)`. Therefore, AdaptFormer can save about 40 times storage with 100 tasks. This advantage can be more significant if the number of different tasks or model parameters becomes larger, which is happening in the current research community.

---

> > ### Author Response · Authors · 2022-08-02
> > **Response to Reviewer 9rzH (Part 2/2)**
> >
> >
> > **5. Evaluation on ImageNet-1K dataset**
> >
> > Thanks for the valuable suggestions. Since we aim to evaluate the fine-tuning performances on datasets different from the pretraining stage and the ImageNet-1K dataset is a subset of the ImageNet-21K [a], we didn't fine-tune models on ImageNet-1K when using ImageNet-21k pre-trained weights.
> >
> > To further validate the effectiveness of our methods, we directly load the weights pre-trained on ImageNet-21K, and evaluate the AdaptFormer on ImageNet-1k. The results in the following table show that our method surpasses the linear probe and VPT in top1 accuracy. Besides, our AdaptFormer achieves comparable performance to the full fine-tuning strategies with only 1.5% parameters.
> >
> >
> > | Method      | Params (M) | Top1   |
> > | ----------- | ---------- | ------ |
> > | Full        | 86.57      |  82.26 |
> > | Linear      | 0.77       | 80.95  |
> > | VPT         | 0.78       | 81.68  |
> > | AdaptFormer | 1.96       | 81.86  |
> >
> >
> > **6. Update of Checklist section**
> >
> > We have discussed the limitations and potential negative societal impacts in the **Conclusion** section rather than in supplemental materials. We have updated the checklist section. Thanks for pointing this out.
> >
> > ---
> > ### Reference
> >
> > [a] Ridnik, Tal, et al. "Imagenet-21k pretraining for the masses." NeurIPS, 2021.

---

> > > ### Comment · Reviewer_9rzH · 2022-08-06
> > > **Responses to authors**
> > >
> > > Dear Authors,
> > >
> > > Thank you for your time and detailed responses. However, I was not convinced on the following two parts:
> > >
> > > 1. The performance difference on ImageNet 1k dataset. As previously pointed out, the reported classification accuracy on Cifar 100 are significantly lower than other paper. The authors address this difference by explaining the settings: the reported accuracy are based on MAE pertained weights instead of the pre-trained weights on ImageNet 1k. However, fine tuning on CIFAR 100 based on ImageNet 1k pertained weights seem to be a more common setting. Besides, MAE also provides fine tuned weights under supervised training on ImageNet 1k. What about the performance on this setting?
> > >
> > > 2.  The experiment results on ImageNet 1k seems that AdaptFormer achieves comparable accuracy with more than 2x tunable parameters than VPT. This trend is different than the other results shown in the paper. How this could be explained?

---

> > > > ### Author Response · Authors · 2022-08-06
> > > > **Response to Reviewer 9rzH (Part 1/2)**
> > > >
> > > > Dear Reviewer 9rzH,
> > > >
> > > > Thanks for your quick reply. We addressed your raised concerns below. We have included all discussions and updated results in our revised manuscript (blue content).
> > > >
> > > > **1. More experiments and discussions on CIFAR-100**
> > > >
> > > > A1: Thank you for your suggestion. We further compared the fine-tuning performance on CIFAR-100 with MAE weights under the supervised training on ImageNet-1k. The results of fine-tuning on CIFAR-100 with different pre-trained checkpoints are summarized in Table 1 below.
> > > >
> > > > **Table 1:** Comparison of fine-tuning performance on CIFAR-100 with different pre-trained weights.
> > > >
> > > >
> > > >
> > > > | Method         | # Params (M) | MAE-SSL Pretrained | MAE Finetuned | Supervised on IN-1K| Supervised on IN-21K |
> > > > | -------------- | ------------ | ------------------ | ------------- | ------------------------ | ------------------------- |
> > > > | Full           | 85.88        | **85.99**          | 87.41         | 86.85                    | 89.12                     |
> > > > | Linear         | 0.08         | 58.74              | 82.18         | 81.43                    | 85.85                     |
> > > > | VPT            | 0.09         | 78.43              | 86.91         | 86.37                    | 90.97                     |
> > > > | AdaptFormer-1  | 0.10         | 81.91              | 87.72         | 86.51                    | 90.54                     |
> > > > | AdaptFormer-4  | 0.16         | 83.86              | 87.90         | 86.76                    | 90.68                     |
> > > > | AdaptFormer-64 | 1.27         | 85.93              | **88.30**     | **87.02**                | **91.86**                 |
> > > >
> > > >
> > > > **Results**
> > > >
> > > > - The performance of all fine-tuning methods on CIFAR-100 varies with different pre-trained weights. For full fine-tuning, MAE fine-tuned weights tend to behave better than MAE self-supervised (SSL) weights (87.41% vs. 85.99%). The highest accuracy with each pre-trained weight is marked in bold.
> > > >
> > > > - We kindly note that the pre-trained ViT-Base model can not easily obtain 90.0%+ accuracy on CIFAR-100 without a specifically designed training regime. For example in ViT [i], ViT-B achieves 87.13% on CIFAR-100 with ImageNet-1K pre-trained weights and 91.67% with ImageNet-21K pre-trained weights（Table 5 in ViT [i]). We achieved 86.85% and 89.12% top-1 accuracy on CIFAR-100 with ImageNet-1K and ImageNet-21K pre-trained weights, respectively. These performances are close to those reported in ViT [i].
> > > >
> > > > - The relatively higher performance of DeiT-B [iii] on CIFAR-100 (90.8%) comes from the specially designed training regime ([Link1](https://github.com/facebookresearch/deit/issues/45#issuecomment-765213622), [Link2](https://github.com/facebookresearch/deit/issues/45#issuecomment-908158938), [Link3](https://github.com/facebookresearch/deit/issues/117#issuecomment-926641538)), which includes 1000 training epochs and more regularizations including cutmix, mixup, RandAugmentation, etc. This setting is carefully designed for full-finetuning on CIFAR-100. In contrast, our AdaptFormer works similarly to the linear probe where only a few parameters are updated. Therefore, we adopt the linear probe settings used in MAE [v], which disabled these regularizations since it is illustrated to be harmful to the liner probe [iv, v].
> > > > - We note that for experiments in the above table, we use the same training setting for apple-to-apple comparisons.

---

> > > > > ### Author Response · Authors · 2022-08-06
> > > > > **Response to Reviewer 9rzH (Part 2/2)**
> > > > >
> > > > > **2. Analysis of the experiments on ImageNet 1K**
> > > > >
> > > > > A2. We kindly point out that in order to evaluate the adaptation performance across datasets, it's an unreasonable setting to fine-tune the ImageNet-1k dataset with the ImageNet-21k pre-trained weights. This is because **ImageNet-1K** is a subset of the **ImageNet-21K** as introduced in [vi]. In contrast, in all the experiments in our paper, there is no overlap between the fine-tuning and pre-trained datasets. However, we would like to thank your suggestions on fine-tuning on the ImageNet-1k dataset for the completeness of our experiments.
> > > > >
> > > > > The number of AdaptFormer parameters could be further reduced since it is controlled by the `middle dimension` of the introduced branch. In our initial rebuttal feedback, we only evaluate AdaptFormer with `middle dimension = 64` on ImageNet-1k dataset. For further exploration, we adopt exactly identical training configurations to conduct experiments with `middle dimension = {1, 4, 16, 64}` on ImageNet-1K, and the results are shown in below table.
> > > > >
> > > > >
> > > > > | Method         | Params (M) | Top1  |
> > > > > | -------------- | ---------- | ----- |
> > > > > | Full           | 86.57      | 82.26 |
> > > > > | Linear         | 0.77       | 80.95 |
> > > > > | VPT            | 0.78       | 81.68 |
> > > > > | AdaptFormer-1  | 0.80       | **82.33** |
> > > > > | AdaptFormer-4  | 0.85       | 82.26 |
> > > > > | AdaptFormer-16 | 1.07       | 82.24 |
> > > > > | AdaptFormer-64 | 1.96       | 81.86 |
> > > > >
> > > > >
> > > > >
> > > > > **Results:**
> > > > >
> > > > > - **(1).** Comparing the results of AdaptFormer with different  `middle dimension` (`{1, 4, 16, 64}`) on ImageNet-1K, we find that AdaptFormer with the smallest number of parameters (***AdaptFormer-1***) achieves the best top-1 accuracy (82.33%).
> > > > > - **(2).** When the `middle dimension` increases from 1 to 4 or 16, AdaptFormer has a slight performance drop (AdaptFormer-4 (-0.07%) and AdaptFormer-16 (-0.09%)). Further increasing the `middle dimension` to 64 will cause a relatively clear performance drop (-0.47%).
> > > > >
> > > > > **Discussions:**
> > > > > - **(1).** Although our previously used AdaptFormer-64 does not have a clear advantage compared with VPT [i], our AdaptFormer-1 outperforms VPT by +0.65% top-1 accuracy with only 0.02M additional parameters.
> > > > >
> > > > > - **(2).** The trend of *classification accuracy* changing with `middle dimension` on ImageNet-1k is different from other datasets in our paper, e.g.,  AdaptFormer with `middle dimension=64` achieves better top-1 accuracy than with `middle dimension=1` on CIFAR-100. We empirically find introducing a small number of parameters (AdaptFormer-1) is sufficient for ImageNet-1K fine-tuning, while introducing more parameters will make a larger change to the original model. This change is harder for optimization since  ImageNet-1K is a subset of **ImageNet-21K**. However, for other datasets (e.g., CIFAR-100, SSv2, etc) with no overlap between the fine-tuning datasets and the pre-trained datasets, more parameters (i.e., `dim=64`) are needed for better domain transfer.
> > > > >
> > > > > Thanks for your raised concern which helps us improve the clarity of this work. We have added these updated results in **Appendix A.5** Evaluation on ImageNet-1k datasets.
> > > > >
> > > > >
> > > > > Thanks again for your time. Wish you have a nice weekend.
> > > > >
> > > > > Best,
> > > > >
> > > > > Authors
> > > > >
> > > > >
> > > > > ---
> > > > >
> > > > > ### Reference
> > > > >
> > > > >
> > > > > [i] Dosovitskiy, Alexey, et al. "An image is worth 16x16 words: Transformers for image recognition at scale." ICLR, 2021.
> > > > >
> > > > > [ii] Jia, Menglin, et al. "Visual prompt tuning." ECCV, 2022.
> > > > >
> > > > > [iii] Touvron, Hugo, et al. "Training data-efficient image transformers & distillation through attention." ICML, 2021.
> > > > >
> > > > > [iv] Chen, Xinlei, et al. "An empirical study of training self-supervised vision transformers." ICCV, 2021.
> > > > >
> > > > > [v] He, Kaiming, et al. "Masked autoencoders are scalable vision learners." CVPR, 2022.
> > > > >
> > > > > [vi] Ridnik, Tal, et al. “Imagenet-21k pretraining for the masses.” NeurIPS, 2021.

---

### Official Review · Reviewer_T4q6 · 2022-07-15

**Rating:** 6
**Confidence:** 4
**Soundness:** 3 good
**Presentation:** 3 good
**Contribution:** 2 fair

**Summary:**

This paper proposes an efficient adaptation approach for vision Transformer (ViT) that enables downstream tasks with limited parameters to tune. Specifically, the MLP in ViT is replaced by a new AdaptMLP, which keeps the original MLP branch but introduces a simple bottleneck branch for tuning. During the finetuning stage, the original MLP, multi-head self-attention and layernorms are frozen. In contrast, the weights of the additional branch in AdaptMLP and the last linear layer for classification are updated. Extensive experiments on both image and video recognition tasks indicate that the proposed approach achieves on-par or even better performance compared with a full-tuning setting and a significant improvement over the prior arts such as VPT.

**Questions:**

See Weaknesses for most questions.

I have an additional question/suggestion here: In AdaptMLP, the MLP and the bottleneck module are parallel. Is it possible to combine them into a single MLP (e.g., unifying the activation function and then merging the weights)?  It would be more efficient during the inference in downstream tasks since GPUs/TPUs are more optimized for a single branch rather than parallel branches.

**Limitations:**

The authors have discussed limitations (e.g., the ability to extend to tasks beyond recognition) and claimed no potential societal impact.

**Strengths And Weaknesses:**


Strengths:
1. This paper proposes a lightweight bottleneck branch to enhance the MLP in ViTs for efficient task transfer. The proposed approaches reuse (almost) all weights from pretrained ViTs that maximize the utilization of pretraining.
2. The proposed AdaptMLP is simple and can be easily applied to multiple variants of ViTs since the MLP module is usually kept intact in many ViT variants.
3. Empirical experiments show that the proposed AdaptFormer is efficient and effective: On image and video recognition tasks, the AdaptFormer obtains way better performance than VPT and linear probing. With less than 2% extra parameters, the AdaptFormer achieves on-par or even better performance than the full-tuning method, which requires all weights to be updated in finetuning.
4. Ablation study shows that the hyperparameter for the AdaptMLP (e.g., bottleneck middle dimension) is not very sensitive to the downstream tasks. It reduces the burden of engineering work on finding good combinations of hyperparameters in pretraining and finetuning.
5. The paper is well written and easy to understand.

Weaknesses:
1. The efficiency of the proposed method is mainly measured in parameters throughout the manuscript. However, it remains unclear how much computation this method introduces in finetuning and inference. Thus, it would be better to include comparisons for finetuning time and additional latency during inference.
2. Line 289 mentions that the accuracy reaches saturation when middle dimension = 64. Is the same middle dimension used for all tasks, or each task still has its own middle dimension? I am curious about this because the ablation study claims that the performance is insensitive to the (proper) choice of the middle dimension.
3. In line 216 Initialization of AdaptFormer, it said that "the weights of down-projection layers are initialized with Kaiming Normal [33], while the biases of the additional networks and the weights of the up-projection layers are configured with zero initialization." I wonder if there is any reason behind such choice of initializations.
4. Table 2(c) scaling factors s is not well analyzed in the paper.
5. Other minor issues: Line 308-309 ourperforms -> outperforms, methhod -> method

---

> ### Author Response · Authors · 2022-08-02
> **Response to Reviewer T4q6 (Part 1/2)**
>
>
> Dear Reviewer T4q6,
>
>
>
> Thank you for appreciating our work and your valuable comments. We answer the raised questions below. We have included all discussions and results in our revised manuscript (blue content in the main text and appendix).
>
>
> **Q1: Comparisons for finetuning time and latency during inference.**
>
> A1: Thank you for the suggestion. We compare the finetuning time and inference time on a single NVIDIA A100-40G GPU. We utilize SSv2 video classification for this part. For fine-tuning, we experiment with `batch size=32`. For inference, we test the latency with multiple batch sizes to get a comprehensive comparison under various inference scenarios. All the time is measured in milliseconds averaged over 100 trials. The results are summarized in Table 1 and 2.
>
> **Table 1.** Fine-tuning time of a single forward-backward step averaged over 100 trials.
>
> | method      | Latency (B=32) |
> | ----------- | -------------- |
> | Full-tuning | 355.0 ms       |
> | Linear      | 140.2 ms       |
> | VPT         | 210.3 ms       |
> | AdaptFormer | 162.2 ms       |
>
>
> **Table 2.** Inference time of a single forward step averaged over 100 trials.
>
> | method             | FLOPs (B=1)  | Latency (B=1) | Latency (B=16) | Latency (B=32) |
> | ------------------ | ------- | ------------- | -------------- | -------------- |
> | Linear/Full-tuning | 78.915G | 11.1 ms       | 22.4 ms        | 42.3 ms        |
> | VPT                | 79.029G | 11.3 ms       | 22.9 ms        | 42.4 ms        |
> | AdaptFormer        | 79.840G | 11.9 ms       | 23.2 ms        | 42.8 ms        |
>
>
> **Discussion:** As shown in Table 1, AdaptFormer only costs less than half of the fine-tuning time compared with the full-tuning. Moreover, AdaptFormer significantly outperforms linear probing in terms of accuracy with a slightly longer fine-tuning time. For inference, AdaptFormer introduces negligible FLOPs and latency compared with the Linear/Full-tuning.
>
>
>
> **Q2: Ablation on middle dimension.**
>
>
> A2: We further conduct more extensive ablation studies on the middle dimension. We evaluate AdaptFormer on video dataset SSv2 and image dataset NUS-WIDE ( a more diverse multi-label dataset suggested by Reviewer KxCH) with middle dimensions {1, 4, 16, 32, 64, 128, 256, 512}.
>
> Results are summarized in the below table. The optimal middle dimension varies per dataset. For example, the accuracy reaches saturation when the `middle dimension=64` on SSv2, whereas for NUS-WIDE dataset, the mAP slightly improves when the middle dimension increases from 64 to 512. However, AdaptFormer with `middle dimension=512` has 0.75 mAP higher (59.82 vs. 59.07 mAP) than the one with `middle dimension=64` at the cost of about 8 times more parameters. Therefore, we choose the `middle dimension=64` for both SSv2 and NUS-WIDE for a better trade-off.
>
>
>
> | Middle Dim | #Params SSv2 | Top1 Acc SSv2 | #Params NUS-WIDE | mAP NUS-WIDE |
> | ---------- | ------------ | ------------- | ---------------- | ------------ |
> | 1          | 0.16         | 50.03         | 0.09             | 57.51        |
> | 4          | 0.22         | 54.70         | 0.15             | 58.14        |
> | 16         | 0.44         | 57.62         | 0.37             | 59.00        |
> | 32         | 0.73         | 58.27         | 0.66             | 59.09        |
> | 64         | 1.32         | **59.02**     | 1.25             | 59.07        |
> | 128        | 2.51         | 58.95         | 2.43             | 59.49        |
> | 256        | 4.87         | 58.87         | 4.79             | 59.62        |
> | 512        | 9.59         | 58.98         | 9.51             | **59.82**    |
>
>
>
> **Q3: Motivation of the initialization method.**
>
>
>
> A3: The Kaiming initialization for the down-projection is a popular initialization method [a]. The reason for the zero initialization of other layers is that in this way, the initial newly added parameters are initialized such that the new function resembles the original one at the start of the fine-tuning stage. We empirically found that if the initialization deviates too far from the identity function, the model is not stable to train.
>
>
>
> **Q4: Discussion of the scaling factor.**
>
>
>
> A4: Thank you for pointing it out. The scaling factor `s` is introduced to balance the *task-agnostic* features (generated by the original frozen branch) and the *task-specific* features (generated by the tunable bottleneck branch). We evaluate AdaptFormer with multiple `s` values, and the results are summarized in Table 2.c (main paper). Different from the scaling factor in the NLP field, which prefers `s` larger than 1 (e.g., `s=4` in [b]), we empirically found that the `s` should be `<1` for vision tasks. Otherwise, the fine-tuning would become unstable. Besides, we discovered that AdaptFormer achieves optimal performance with `s=0.1`. A larger or smaller `s` would bring a slight performance drop. Therefore, we choose `s=0.10` as our default setting.

---

> > ### Author Response · Authors · 2022-08-02
> > **Response to Reviewer T4q6 (Part 2/2)**
> >
> >
> > **Q5: Typos.**
> >
> >
> >
> > A5: Thank you for pointing out the typos. We have carefully revised the manuscript and corrected the typos in the updated manuscript.
> >
> >
> >
> > **Q6: Promising direction to merge two branches into a single MLP.**
> >
> >
> >
> > A6: Thanks for this fruitful suggestion. We agree that merging two parallel branches into a single branch would bring benefits to the modern AI hardware (e.g., GPU and TPU). Recent work [c] has demonstrated that it is a meaningful research topic. We will explore this direction in our future work.
> >
> >
> >
> > We would like to express our appreciation again for your constructive suggestions, which help improve this paper.
> >
> >
> >
> > ---
> >
> > ### Reference
> >
> >
> >
> > [a] He, Kaiming, et al. "Deep residual learning for image recognition." CVPR. 2016.
> >
> > [b] He, Junxian, et al. "Towards a unified view of parameter-efficient transfer learning." ICLR. 2022.
> >
> > [c] Ding, Xiaohan, et al. "Repvgg: Making vgg-style convnets great again." CVPR. 2021.

---

> > > ### Comment · Reviewer_T4q6 · 2022-08-09
> > > **Reply to authors**
> > >
> > > Thank you for your detailed reply! I think all my questions are well addressed. I would keep my accept rating.

---

### Official Review · Reviewer_fSbd · 2022-07-23

**Rating:** 5
**Confidence:** 4
**Soundness:** 3 good
**Presentation:** 3 good
**Contribution:** 2 fair

**Summary:**

This paper explores leveraging adapter in vision transformers. Similar to adapter-transformer in NLP, this is done by replacing the MLP layer in each transformer block with a adapter-MLP module. In finetuning, only a small number of parameters in the adapter-MLP is updated.  For image classification datasets, the proposed method AdaptFormer is on par or of lower performance than fully-finetuned model. For video datasets, where training set is usually in smaller scale than image classification datasets, this proposed parameter-efficient tuning technique achieves good performance, outperforming fully-funetuned models. Authors also conduct ablation study, e.g., the effect of #layers and the size of intermediate dimension of adapter-MLP module on the performance.

**Questions:**

Have you tested on ImageNet dataset, as in the original ViT paper?

**Ethics Review Area:**

["Discrimination / Bias / Fairness Concerns", "Inadequate Data and Algorithm Evaluation", "Inappropriate Potential Applications & Impact  (e.g., human rights concerns)", "Privacy and Security (e.g., consent)", "Legal Compliance (e.g., GDPR, copyright, terms of use)", "Research Integrity Issues (e.g., plagiarism)", "Responsible Research Practice (e.g., IRB, documentation, research ethics)", "I don’t know"]

**Limitations:**

Authors discussed it briefly in the conclusion section.

**Strengths And Weaknesses:**

Strengths:
The proposed methodology is clearly presented. The ablation study and exploration is comprehensive. The parameter-efficient performance on video dataset is impressive.

Weakness:
1. In my opinion, the contribution of the proposed method is incremental. The adapter architecture is essentially the same as the one in the adapter paper [1]. Applying adapter in vision-related tasks, e.g., VQA and vision-grounding, have been explored in [2] and [3]. Actually, many different parameter-efficient tuning techniques other than adapter, e.g., Hyperformer and Compacter, have also been explored in these previous works.

2. The adapter module proposed in [1] is originally designed for NLP. Considering the spatial and channel-wise properties of vision inputs, I'm wondering have authors ever considered other adapter architectures that can be potentially more suitable for vision modeling?

Reference:
[1] Parameter-Efficient Transfer Learning for NLP, 2019
[2] VL-ADAPTER: Parameter-Efficient Transfer Learning for Vision-and-Language Tasks, CVPR
[3] Adaptive Fine-tuning for Vision and Language Pre-trained Models, NeurIPS

---

> ### Author Response · Authors · 2022-08-02
> **Response to Reviewer fSbd (Part 1/2)**
>
>
> Dear Reviewer fSbd,
>
> Thank you for your comments. We answer the raised questions below. We have included all discussions and results in our revised manuscript (blue content).
>
> **1. Contributions recap.**
>
> We provide a conceptual comparison between previous methods (i.e., Adapter [a], VL-ADAPTER [b], AFVL [c]) and ours in the appendix in the revised manuscript. The relationship between ours and existing methods is summarized as follows:
>
> #### Similarity
>
> We agree that there are some similarities between the previous methods (i.e., Adapter[a], VL-ADAPTER [b], AFVL[c]) and our work. Specifically, both the previous methods and our AdaptFormer achieve efficient tuning via introducing external networks. Meanwhile, the MLP architectures are widely utilized among the above methods.
>
> #### Differences
>
> - The target data modalities needed to be adapted are different. Previous works [a, b, c] focus on processing natural language data. Even though VL-ADAPTER [b] and AFVL [c] introduce the fine-tuning strategy in vision-related tasks, e.g., VQA and vision-grounding, they only apply the adapter to process the language data, leaving the vision data unprocessed. Different from previous works, our AdaptFormer is the first to apply the adapter to the computer vision field and address the scalable recognition problem (i.e., both image and video data).
>
> - The number of parameters introduced to the fine-tuning process is different. The tunable parameters make for about 40 percent of the one in full fine-tuning strategy in AFVL [c], and about 5 percent in VL-ADAPTER [b]. Unlike the above methods, the updated parameters only cover less than 2 percent of the full fine-tuning one.
>
> - The way to employ the adapter architecture is different. Previous works [b, c] mainly introduce the adapter architectures in a sequential fashion to the original networks, while our AdaptFormer empirically finds that the parallel fashion is the optimal way for vision tasks (detailed discussion can be found in Sec 4.4: **Insertion form**).
>
> **2. Other adapter architectures**
>
> Thanks for your suggestion. We design some extensions of our AdaptFormer by taking the spatial and channel-wise properties of visual inputs. Specifically,
> - **(1.)** We insert a 2D convolution layer with `kernel size=3` and `stride=1` after the non-linear function of our introduced bottleneck structure;
> - **(2.)** Similar to **(1.)**, we replace the vanilla 2D convolution with a depthwise convolution layer, reducing the number of parameters and FLOPs of the convolution layer;
> - **(3.)** We add a LayerNorm to the input of the newly introduced AdaptMLP branch;
>
> We comprehensively compare these structure variants on multiple tasks, including SSv2 video classification, NUS-WIDE image multi-label classification, and CIFAR100 image classification. We have included the results in Table 8 in the appendix.
>
> | method         | Avg Params | SSv2 Top1 | NUS-WIDE mAP | CIFAR100 Top1 |
> | -------------- | ---------- | --------- | ------------ | ------------- |
> | Conv           | 1.39       | 58.47     | 58.8573      | 85.42         |
> | Depthwise Conv | 1.29       | 58.15     | 58.7262      | 85.37         |
> | LayerNorm-In   | 1.30       | 57.85     | 58.5123      | 85.71         |
> | AdaptFormer    | 1.28       | 59.02     | 59.0663      | 85.93         |
>
>
> **Discussion:** As shown in the above table, our AdaptFormer consistently outperforms other variants on these three benchmarks. Therefore, we recommend our default simple yet effective AdaptFormer design.

---

> > ### Author Response · Authors · 2022-08-06
> > **Response to Reviewer fSbd (Part 2/2)**
> >
> >
> > **3.Evaluation on ImageNet-1K dataset**
> > We kindly point out that in order to evaluate the adaptation performance across datasets, it's an unreasonable setting to fine-tune the ImageNet-1k dataset with the ImageNet-21k pre-trained weights. This is because **ImageNet-1K** is a subset of the **ImageNet-21K** as introduced in [d]. In contrast, in all the experiments in our paper, there is no overlap between the fine-tuning and pre-trained datasets. However, we would like to thank your suggestions on fine-tuning on the ImageNet-1k dataset for the completeness of our experiments.
> >
> >
> > We adopt exactly identical training configurations to conduct experiments with `middle dimension = {1, 4, 16, 64}` on ImageNet-1K, and the results are shown in the below table.
> >
> >
> > | Method         | Params (M) | Top1  |
> > | -------------- | ---------- | ----- |
> > | Full           | 86.57      | 82.26 |
> > | Linear         | 0.77       | 80.95 |
> > | VPT            | 0.78       | 81.68 |
> > | AdaptFormer-1  | 0.80       | **82.33** |
> > | AdaptFormer-4  | 0.85       | 82.26 |
> > | AdaptFormer-16 | 1.07       | 82.24 |
> > | AdaptFormer-64 | 1.96       | 81.86 |
> >
> > **Results:**
> >
> > - **(1).** Comparing the results of AdaptFormer with different  `middle dimension` (`{1, 4, 16, 64}`) on ImageNet-1K, we find that AdaptFormer with the smallest number of parameters (***AdaptFormer-1***) achieves the best top-1 accuracy (82.33%).
> > - **(2).** When the `middle dimension` increases from 1 to 4 or 16, AdaptFormer has a slight performance drop (AdaptFormer-4 (-0.07%) and AdaptFormer-16 (-0.09%)). Further increasing the `middle dimension` to 64 will cause a relatively clear performance drop (-0.47%).
> >
> > **Discussions:**
> > The trend of *classification accuracy* changing with `middle dimension` on ImageNet-1k is different from other datasets in our paper, e.g.,  AdaptFormer with `middle dimension=64` achieves better top-1 accuracy than with `middle dimension=1` on CIFAR-100. We empirically find introducing a small number of parameters (AdaptFormer-1) is sufficient for ImageNet-1K fine-tuning, while introducing more parameters will make a larger change to the original model. This change is harder for optimization since  ImageNet-1K is a subset of **ImageNet-21K**. However, for other datasets (e.g., CIFAR-100, SSv2, etc) with no overlap between the fine-tuning datasets and the pre-trained datasets, more parameters (i.e., `dim=64`) are needed for better domain transfer. Moreover, our AdaptFormer-1 outperforms all other fine-tuning methods on ImageNet-1K. In particular, AdaptFormer outperforms VPT by +0.65% top-1 accuracy with only 0.02M additional parameters.
> >
> > Thanks for your raised concern which helps us improve the clarity of this work. We have added these updated results in **Appendix A.5** Evaluation on ImageNet-1k datasets.
> >
> > ---
> > ### Reference
> >
> > [a] Houlsby, Neil, et al. "Parameter-efficient transfer learning for NLP." ICML 2019.
> >
> > [b] Sung, Yi-Lin, et al. "Vl-adapter: Parameter-efficient transfer learning for vision-and-language tasks." CVPR 2022.
> >
> > [c] hentong Mo, et al. "Adaptive fine-tuning for vision and language pre-trained models." NeurIPS Workshop 2021.
> >
> > [d] Ridnik, Tal, et al. “Imagenet-21k pretraining for the masses.” NeurIPS 2021.

---

> > > ### Comment · Area_Chair_zrWw · 2022-08-08
> > > **Review Reminder: Please respond to author's rebuttal.**
> > >
> > > Dear Reviewer, please remember to reply to the author's rebuttal. Thank you.

---

> > > ### Comment · Reviewer_fSbd · 2022-08-08
> > > **Reply to Authors**
> > >
> > > Thank authors for the reply. It's great to see the other parameter-efficient finetuning architectures are also explored. Thank you for the experiments on ImageNet-1K. I've raised the score to 5.

---

### Author Response · Authors · 2022-08-02
**Summary of our rebuttal and discussion**

Dear Reviewers and ACs:


We sincerely appreciate all reviewers for their time and efforts in reviewing our paper. We are glad to find that reviewers recognized the following contributions of our work:


-  **Important topic and much valuable in the research community:** This well-motivated paper introduced a good direction to maximize parameter sharing among multiple tasks. Adaptation of large-scale pre-trained models is meaningful in the real-world application and much valuable in the research community, making the recent popular large-scale pre-trained Transformer models more useful. [9rzH, KxCH]

- **Efficient and effective method demonstrated by extensive experiments with promising results:** Proposing a simple yet effective AdaptMLP module that can be easily applied to multiple variants of ViTs [T4q6]. Extensive experiments and promising results demonstrate the superiority of AdaptFormer, especially in the video domain.[fSbd, T4q6, 9rzH]. Ablation studies and exploration are comprehensive.

- **Well-written paper and clearly-presented methodology:** The paper is well written and easy to understand. [fSbd, T4q6, 9rzH, KxCH].


We also thank all reviewers for their insightful and constructive suggestions, which help further improve our paper. In addition to the pointwise responses below, we summarize the major revision in the rebuttal according to the reviewers’ suggestions.

* **Extended Experiments:** ([fSbd, T4q6, 9rzH, KxCH]) We evaluate AdaptFormer on more diverse image dataset NUS-WIDE (**Appendix A.3 Multi-Label Classification**) and ImageNet-1K (**Appendix A.5 Evaluation on ImageNet-1k datasets**). Besides, we also conduct experiments with some structure variants of our AdaptFormer (**Appendix A.4 Possible Architectures**). In addition, we comprehensively analyze the middle dimension (**A.6 Extended experiments on middle dimension**), computational cost, and memory usage (**Appendix A.7 Analysis on the fine-tuning time and inference latency**).

* **Code:** We update the code in the supplementary materials, including more experiments discussed in the rebuttal period.

* **Manuscript update**: We clarify t-SNE visualization([KxCH]) and add discussions about the scaling factor. We also add more experimental results and discussions in the Appendix (**A3-A8**). The updated contents are included in our revision of the paper (content in blue).


We really thank all reviewers’ and ACs’ time and efforts again.

Best wishes,

Authors

---

### Author Response · Authors · 2022-08-06
**Sincerely Look Forward to Your Feedback**

Dear AC and all reviewers:

Thanks again for all of your constructive comments and suggestions, which have helped us improve the quality and clarity of this paper!

We sincerely hope that our added experiments and analyses could address your concerns.

Since the deadline for discussion is approaching, please feel free to let us know if there are any additional clarifications or experiments that we can offer, as we would love to convince you of the merits of our work. We appreciate your suggestions.



Best wishes,

Authors

---

### Meta-Review · Area_Chair_zrWw · 2022-08-20

**Recommendation:** Accept
**Confidence:** Less certain

**Metareview:**

Authors introduce lightweight parallel FC layers to the MLP layers for fine-tuning, freezing all other parameters of the model. Experiments across 5 datasets (3 Image: CIFAR-100, SVHN, Food-101 -- 2 Video: SSv2, HMDB51) show that performance is similar to full-finetuning while changing less than 2% of the parameters. Some performance improvements are noted in the video domain.

Pros:
- [R/AC] Paper is well written
- [R/AC] Ablation study is comprehensive
- [R/AC] Performance is impressive
- [R/AC] Efficiency gains for finetuning are significant.
- [R/AC] Modifications are simple and can be applied to variety of ViT architectures.

Cons:
- [R/AC] The novelty is low. In regards to its motivation, the work shares a great deal of similarity to LORA. In regards to its technical approach, only minor adjustments are made in comparison to LORA, with the studied application domain being vision.
- [R/AC] Authors do not provide sufficiently clear description on how learning parameters were chosen across AdaptFormer and full fine-tune approaches. Were parameters independently optimized, or fixed?
- [AC] Method seems very sensitive to weight initialization and scaling mixing parameter s.
- [R] Efficiency was mainly measured as parameters. More information regarding latency and FLOPS should be added. Authors have addressed this concern by including additional information, and should ensure that this new information is in the final manuscript.
- [R] Motivation behind different weight initialization schemes is not clear. Authors have sufficiently answered with more information.
- [R] Need more analysis on why 64 dimensions is the optimal point. What is the explanation? Authors didn't really answer this question other than providing an ablation. Insight behind result is not clear.
- [R] Evaluation is not thorough enough. Some of the datasets used are saturated. Authors added one dataset (NUS-WIDE) to appendix.
- [R] Rationale of design is unclear. [AC] This approach is from a class of similar approaches previously studied. Reviewers also answered this part well.


Reviewer consensus is that this work in its current form is borderline, but the majority of reviewers lean toward accept. AC has concerns about novelty of the method and missing details about how finetuning parameters were chosen across the various experiments. AC will maintain reviewer tendency toward accept, though will comment that the work can be improved if full details about how fine-tuning parameters were chosen across experiments, and that authors optimized those parameters independently in each experiment.

AC Rating: Borderline Accept

**Award:**

No

---

### Decision · Program_Chairs · 2022-09-14

Accept